# Time-Dependent Seismic Reliability of Coastal Bridge Piers Subjected to Nonuniform Corrosion

**DOI:** 10.3390/ma16031029

**Published:** 2023-01-23

**Authors:** Wenting Yuan, Xiangtong Wu, Yuren Wang, Zhenliang Liu, Peng Zhou

**Affiliations:** 1Institute of Transportation, Inner Mongolia University, Hohhot 010070, China; 2State Key Laboratory for Geomechanics & Deep Underground Engineering, School of Mechanics and Civil Engineering, China University of Mining and Technology, Xuzhou 221116, China; 3Ministry-of-Education Key Laboratory of Structural Dynamic Behavior and Control, School of Civil Engineering, Harbin Institute of Technology, Harbin 150090, China; 4Key Laboratory of Large Structure Health Monitoring and Control, School of Safety Engineering and Emergency Management, Shijiazhuang Tiedao University, Shijiazhuang 050043, China

**Keywords:** coastal bridge piers, nonuniform corrosion, moment resistance distribution, moment demand, time-dependent reliability

## Abstract

Coastal bridge piers suffer random performance deterioration owing to the presence of complex nonuniform corrosion characteristics and material uncertainties. Some of these piers will also be threatened by random earthquakes during a long-term service period, and therefore, structural safety needs to be probabilistically assessed by the seismic reliability method. To deal with this problem, we present a method to calculate the time-dependent reliability of the coastal bridge pier, comprehensively considering the randomness of a seismic event, nonuniform corrosion, and material uncertainty. First, the time-dependent M–N interaction diagrams are established by using the Monte Carlo simulation method. On the basis of the interaction diagrams, the moment resistance reduction function and time-dependent moment resistance distribution are determined. Subsequently, the moment demand under the seismic load is determined using the Poisson model and the response acceleration spectrum. Then, the formulas to calculate the time-dependent reliability of a nonuniform corroded pier are derived on the basis of the theorem of total probability. The proposed method is illustrated with a case study of a coastal bridge pier. It was found that the increase in corrosion damage would obviously increase time-dependent reliability. Furthermore, the increase in submerged zone height delayed the year when the failure section shifts from the pier bottom to the bottom of the splash and tidal zone, and it reduces the failure probability of the coastal pier. The research results presented herein show that the nonuniform corrosion manifestations influence the failure mode–related time-dependent seismic reliability of the coastal bridge pier.

## 1. Introduction

Coastal bridges are important parts of the rapid traffic network in coastal regions. However, these bridges are exposed to particularly aggressive chloride-induced corrosion under the condition of poor-quality concrete or/and inadequate cover depth. Once corrosion has been triggered, the mechanical properties and geometrical dimensions of reinforcement and concrete have deteriorated [1,2], leading to long-term structural performance deterioration. This kind of performance degradation should not be assessed by the deterministic method, because of the presence of material and corrosion environment uncertainties. As an alternative, the probabilistic structural performance assessment method expressed by time-dependent reliability or failure probability has gained increasing prominence [3,4,5,6].

Extensive studies have been conducted for decades on the reliability of corroded bridge superstructure components, including beams, decks, and girders. Scholars developed models to calculate the possibilities of structural failure for the RC slab bridge and the time-variant reliability of the bridge deck [7,8]. The calculation results indicated that a reduced concrete cover increased the structural failure probability. For corroded girder, some researchers were interested in its time-variant reliability. It was found that the service reliability [9] and flexure reliability index [10] of the post-tension (PT) bridge girder reached a value below the recommended value within a short service time after chlorides and moisture had infiltrated the tendons. Al-Mosawe et al. [11] conducted a time-dependent reliability analysis of the segmental PT bridge that factored in the knife edge load and corrosion of internal tendons in beams. The results made up for the limitation that the internal tendons were assumed to be uncorroded in existing research. Other researchers devoted their efforts to estimate the time-variant reliability of the PSC bridge girder. It was identified that neglecting the pitting corrosion in the reliability analysis reduced both the probability of serviceability and strength failure [12], and the reliability index of the bridge structure was much larger than the reliability indices of an individual girder for both the serviceability limit state and the ultimate limit state [13]. Using the Taylor series expansion and Gaussian numerical integral method, Yuan et al. [14] investigated the time-dependent reliability of concrete bridges influenced by nonstationary vehicle loads and steel corrosion in the T-beam. Luo et al. calculated the time-dependent fatigue reliability of the prestressed concrete bridge on the basis of the stochastic traffic load model and the corrosion-related S-N curve [15]. Liu et al. computed the time-variant reliability of the suspension bridge with the corrosion fatigue of wires of suspenders, and a refined load model was implemented into the wind-vehicle-bridge system [16]. To improve the mechanical performance of a corroded prestressed bridge, Costa et al. [17] strengthened the girder with externally bonded CFRP laminates. The results showed that the reliability reduction is first fast and then slow over the years after rehabilitation. Based on the time-dependent reliability analysis, the life-cycle design framework of the bridge girder was suggested by [18,19], and this framework can be further used by researchers and engineers to design a more resilient bridge structure.

Compared with superstructure components, relevant studies on time-dependent reliability for corroded bridge pier columns are scarce. The pier column is the main element to suffer from gravity load and live loads. Kliukas et al. [20] presented the time-variant reliability of spun bridge columns under gravity and transient loads. It was found that the reliability index of the column decreased with service time owing to corrosion damage. Zhu et al. [21] presented a probabilistic method to calculate the time-dependent reliability of RC bridge components and selected a bridge column as an illustrative example. It was found that the time-dependent reliability was significantly influenced by the water-cement ratio and concrete cover depth. Given that the bridge pier column belonged to an axial-flexural component, Castaldo et al. [22] evaluated the time-dependent axial force and bending-moment resistance curves and estimated the time-varying reliability of the deteriorated bridge pier. The results implied that the bridge pier subjected to high values of bending-moment actions was influenced mainly by the corrosion effects. Pugliese et al. [23] presented a method to calculate the time-dependent reliability of bridge piers by factoring in the coupled effect of the increasing traffic demand and spatially variable pitting corrosion.

For bridge pier columns in earthquake-prone regions, their seismic reliability should be assessed in the structural safety analysis [24,25,26]. Seismic reliability is influenced by corrosion and is therefore time dependent for the long-term serviced bridge column. Aiming to solve this problem, Guo et al. [27] proposed a time-variant corrosion rate model, and this model was incorporated into the time-dependent reliability analysis procedure of a corroded bridge pier subjected to seismic action. It was found that the seismic reliability of an in-service bridge pier significantly reduced over a short time because of corrosion. To assess the life-cycle seismic reliability of corroded RC bridge piers, Akiyama et al. [28] proposed a computational procedure and considered the probabilistic hazard associated with airborne chlorides. The results showed that the cumulative-time failure probabilities of RC bridge columns in seismic zones were significantly influenced by airborne chloride corrosion. To make a more precise prediction of the life-cycle seismic reliability of a corroded bridge pier, [29,30] adopted the X-ray digital-picture-processing method to obtain the spatial distribution of rebar corrosion and incorporated it into the structural model. The analytical results revealed that even if the failure probability is relatively low in the short service period, it will obviously increase with time as a result of corrosion. Asghshahr [31] conducted a sampling-based seismic reliability analysis of a corroded reinforced concrete bridge pier and found that seismic input is the most sensitive parameter on the reliability of the corroded bridge bent. Zanini et al. [32] established the relationship between the time-dependent reliability of the corroded pier and the number of FRCM layers, providing a way to ensure an adequate seismic performance level of aging bridge structures by using composite materials.

Although the above research works have investigated the time-dependent reliability of urban bridge piers subjected to nearly uniform corrosion, the relevant results and conclusions cannot reflect the time evolution of the reliability of the coastal bridge piers, because of their distinct corrosion characteristics. In general, a coastal pier column can be divided into three zones along its height and characterized by nonuniform corrosion, as shown in Figure 1. The splash and tidal zone is the most severe corrosion region, followed by the atmospheric zone, and the corrosion in the submerged zone can be neglected [33]. For this kind of nonuniform corroded coastal bridge pier, when the service time is short, the section in the submerged zone has the lowest reliability because of the largest bending-moment demand along the pier elevation. However, over time, the critical failure section may shift to the splash and tidal zone when the corrosion damage induces serious moment-capacity loss in this zone of the pier. Because of the transfer of the failure section, the failure mode–related reliability of the coastal bridge pier is different from that of the urban bridge pier, whose failure is controlled at the bottom section. Therefore, the time-dependent reliability of coastal bridge piers when suffering random earthquake excitation is different from that of uniform corroded piers. Because obtaining the seismic reliability evolutionary process of the nonuniform corroded coastal pier will aid in the life-cycle assessment of coastal bridges, a study related to this topic is urgently needed.

This study proposes a framework to predict the time-dependent seismic reliability of coastal bridge columns suffering from nonuniform corrosion. The method comprehensively considers the material uncertainty as well as the randomness of a seismic event and nonuniform corrosion. The main contents consist of six sections. The first section presents the method to establish the time-dependent resistance deterioration model of coastal bridge piers. The second section illustrates how to calculate the individual seismic load demand on coastal bridge piers. On the basis of the first and second sections, the formulas to calculate the time-dependent reliability of coastal bridge piers are deduced and presented, in the fourth section. In the fifth section, the proposed time-dependent reliability model is illustrated by a representative case study of a coastal bridge pier. The sixth section presents the comparable previous work on the seismic reliability evaluation of corroded bridge piers, to highlight the limitations of the previous work. Finally, the main conclusions are summarized in the seventh section.

## 2. Time-Dependent Resistance Deterioration of Coastal Bridge Piers

### 2.1. Corrosion-Initiation Time

For the in-service RC structure in the coastal region, chloride ions from seawater penetrate through the concrete cover to the reinforcing steel. After the passivation film dissolves and the chloride ions accumulate to a certain extent, the chloride corrosion of the reinforcing steel is initiated. The corrosion-initiation time *T*_I_ can be estimated by the following [34]:(1)$$T_{I}=\frac{c^{2}}{4D_{c}}{\left[{\mathrm{erf}}^{-1}\left(\frac{C_{0}-C_{cr}}{C_{0}}\right)\right]}^{-2}$$
where *c* is the concrete cover depth; *D*_c_ is the chloride diffusion coefficient; *C*_0_ is the equilibrium chloride concentration at the concrete surface; and *C*_cr_ is the critical chloride concentration.

### 2.2. Deterioration Models of Reinforcement and Concrete

Once corrosion starts, the diameter *D*(*t*) and yield stress *f*_y_(*t*) of the reinforcement are continuously reduced [1,2], which can be expressed as follows:(2)$$D(t)=\left\{\begin{array}{ccc} \begin{array}{l} D_{0}\\ D_{0}-0.023i_{\mathrm{corr}}\cdot (t-T_{I})\\ 0 \end{array} & \begin{array}{l} \mathrm{for}\\ \mathrm{for}\\ \mathrm{for} \end{array} & \begin{array}{l} t\leq T_{I}\\ T_{I}<t\leq T_{I}+D_{0}/(0.023i_{\mathrm{corr}})\\ t>T_{I}+D_{0}/(0.023i_{\mathrm{corr}}) \end{array} \end{array}\right.$$
(3)$$f_{y}(t)=\left[1-0.005\cdot Q_{\mathrm{corr}}(t)\right]f_{y0}$$
where *D*_0_ is the initial diameter of the uncorroded reinforcement; *f*_y0_ is the initial diameter of the uncorroded reinforcement; *i*_corr_ is the corrosion current density, which can be obtained from [35]; and *Q*_corr_(*t*) is the mass loss ratio of corroded reinforcement corresponding to a structural service time *t*, which can be determined as
(4)$$Q_{corr}(t)=[1-{\left(D(t)/D_{0}\right)}^{2}]\times 100$$

During the corrosion process, the volume of the reinforcement gradually increases thanks to the formation of corrosion products. Consequently, the concrete around the reinforcement suffers expansive force, leading to concrete cover damage, including cracking and even spalling. The strength of the damaged concrete cover can be calculated by Equation (5) [36,37]:(5)$$f_{c}(t)=\frac{f_{c0}}{1+K\varepsilon_{1}(t)/\varepsilon_{c0}}$$
where the coefficient *K* is related to the morphology of the reinforcement, and *K* = 0.1 is used in this study based on [33]; *f*_c0_ is the strength of the uncorroded concrete with a peak strain $\varepsilon_{c0}$; and $\varepsilon_{1}(t)$ is the average value of the tensile strain of the corrosion-induced cracked concrete.

### 2.3. Time-Dependent Resistance Distribution under Material Uncertainty and Nonuniform Corrosion

As a result of the chloride-induced damage of reinforcement and concrete, the resistance (load-carrying capacity) of in-service coastal bridge piers deteriorates. For coastal bridge piers in seismic-prone regions, one important aspect for representing the resistance degradation phenomenon can be expressed by the time-dependent M–N interaction diagram, given that piers suffered from axial force and bending moment induced by the gravity and earthquake load. On the basis of the fiber discretization method, and given the corrosion influence with the method in Section 2.2, the time-dependent M–N interaction diagram for a given cross section of the deteriorated bridge pier with bending failure pattern can be written as follows [33]:(6)$$N(t)=F_{c}(t)+\sum_{i=1}^{n_{1}}F_{sci}(t)-\sum_{j=1}^{n_{2}}F_{stj}(t)$$
(7)$$M(t)=M_{c}(t)+\sum_{i=1}^{n_{1}}F_{sci}(t)\cdot [\frac{h(t)}{2}-d_{sci}(t)]-\sum_{j=1}^{n_{2}}F_{stj}(t)\cdot [\frac{h(t)}{2}-d_{stj}(t)]$$
where *N*(*t*) and *M*(*t*) are the axial force and the bending-moment resistance, respectively; *n*_1_ and *n*_2_ are the numbers of the compression and tension reinforcement; *F*_c_(*t*) is the axial force contributed by the concrete; *F*_sci_(*t*) is the axial force contributed by the *i*-th compression reinforcement; and *F*_stj_(*t*) is the axial force contributed by the *j*-th tensile reinforcement. More details on the bending-resistance calculation under any axial force of the corroded pier can be found in the aforementioned literature.

It is well known that the material properties of the reinforcement and of concrete, and the exposure corrosion environment of practical coastal bridges, are random, with strong uncertainty. Consequently, the structural resistance under axial compression and bending also exhibits randomness and uncertainty. To determine this uncertainty, if the probability distribution of random variables affecting the bridge piers for a specific service time is given, several sampling results of the M–N interaction diagrams can be obtained by using the Monte Carlo simulation (MCS) method. Section 5 summarizes the material property and geometrical size of a bridge pier, used as an example. The M–N interaction diagrams at 0 years determined by the Monte Carlo simulation with 10,000 iterations are shown in Figure 2a. By calculating the intersecting point of each M–N interaction diagram and the axial force line (represented by a black dotted line), the bending-moment resistance corresponding to an axial force of 4850 kN can be determined. The statistical analysis results of the bending-moment resistance are presented in Figure 2b. As Figure 2b shows, the moment resistance presents strong randomicity and can be fitted with a lognormal distribution function. The mean moment resistance is 7962 kN⋅m with a standard deviation of 720 kN⋅m. Using the same method, for a given axial force and given service time of the pier, the probability distribution of the bending-moment resistance can be consequently obtained. 

From the viewpoint of the computational cost and to facilitate the reliability analysis in the following subsection of this study, MCS is conducted only at some discrete time points and the time-dependent resistance factoring in randomness is expressed as follows [38]:(8)$$R(t)=R_{0}\cdot g(t)$$
where *R*(*t*) is the computed bending-moment resistance of the corroded bridge pier; *R*_0_ is the moment resistance with initial uncertainty; and *g*(*t*) is the moment resistance deterioration function, which can be determined by Equation (9). The linear, parabolic, or square root function was used to describe *g*(*t*) by [38]:(9)$$g(t_{i})=R_{m}(t_{i})/R_{m0}$$
where *t*_i_ is the discrete time point and *R*_m0_ and *R*_m_(*t*_i_) are mean values of the moment resistance of pier section at time *t* = 0 and *t* = *t*_i_, respectively. Actually, *g*(*t*) is not only time variant but also stochastic thanks to the randomness of the material property and corrosive environment. In this study, the complicated stochastic nature of *g*(*t*) is not considered, because of the insufficient data to quantify the variability in degradation rate, and it will be investigated in a future study. By using Equation (8), when *g*(*t*) and the probabilistic distribution of *R*_0_ is determined and uncertainty is factored in, the time-dependent resistance *R*(*t*) is obtained in a simplified way.

According to the above illustration, it is found that the moment resistance of the pier depends on the sectional material property. For a coastal bridge pier, the time-dependent sectional material property is different among three corroded zones [33]. This nonuniform corrosion characteristic of piers leads to different probabilistic distributions of corrosion-initiation time and reinforcing steel and concrete deterioration. Consequently, there are three probabilistic distributions of sectional moment resistance in three zones of the coastal pier. Additionally, the pier section in the splash and tidal zone suffers more-severe corrosion-induced resistance degradation than that in the atmospheric zone [33], and the resistance deterioration in the submerged zone is neglected because of a lack of oxygen [39]. In this study, the nonuniform deteriorated moment resistance of three zones should be considered and separately calculated in the seismic reliability analysis.

## 3. Seismic Action

The long-term serviced coastal bridge columns may suffer different kinds of natural or humanmade hazards. Among them, a seismic event is one of the most common and destructive hazards for coastal bridge piers in earthquake-prone regions. For cantilever bridge piers, the seismic action can be represented by an equivalent inertia force acting on the pier top, as shown in Figure 3. Because of the uncertainty of the occurrence time and the intensity of the seismic events, the bending-moment demand of the bridge pier also exhibits strong randomness.

Compared with the service period of a structure, the duration of a seismic event is generally very short. Hence, over the whole service period, the random sectional moment demand *S* acting on the pier can be modeled as a sequence of randomly occurring pulses with random intensity $S_{1},S_{2},...,S_{n}$ with the duration τ [38]. For coastal bridge piers, the seismic demand pulse sequences at three zones are shown in Figure 3. As presented in the figure, the bending-moment demand *S* and the moment resistance *R* (determined from Section 2) at three zones are different at the same time instance. Because of the sufficiently short seismic event duration τ, the resistance of the structure during this duration is regarded as constant.

Then, the seismic event can be described by a Poisson point process, and the probability of the seismic load event *N*(*t*_L_) occurring within the time interval $\left(0,t_{L}\right]$ can be expressed by [38]:(10)$$P[N(t_{L})=n]=\frac{{\left(\lambda t_{L}\right)}^{n}}{n!}e^{-\lambda t_{L}};n=0,1,2,...$$
where *P*[] is the probability of the seismic event in the bracket and λ is the mean occurrence rate of the seismic event. Using the Poisson model, the probability distribution of the maximum seismic load effect occurs during a reference period *T* with a time interval of $\left(0,t_{L}\right]$, which can be calculated with [40], as follows:(11)$$F_{S\max}(s,T)=\sum_{n=0}^{\infty }P{\left[S_{max}<s|N(T)=n\right]}^{n}\cdot P\left[N(T)=n\right]=\exp\left[-\lambda T\cdot (1-F_{S}(s))\right]$$
where *F*_Smax_ is the cumulative distribution function of the maximum seismic effect and *F*_S_(*s*) is the cumulative distribution function of the individual seismic pulse effect.

On the other hand, the maximum equivalent seismic load *F*_equal_ during a reference period *T* (*T* = 50 years) can be calculated by
(12)$$F_{\mathrm{equal}}=W\cdot S_{a}$$
where the superstructure weight *W* acting on the pier top is treated as constant because *W* is less random than the seismic intensity. *S*_a_ is the spectral acceleration. According to the NCHRP response spectrum [41], *S*_a_ is a function of the natural periods *T*_N_ of the bridge pier (see Figure 4) and can be described by Equation (13):
(13)$$S_{a}=\left\{\begin{array}{cc} 0.6\frac{S_{DS}}{T_{0}}T_{N}+0.4S_{DS} & T_{N}<T_{0}\\ S_{DS} & T_{0}<T_{N}<T_{s}\\ \frac{S_{D1}}{T_{N}} & T_{N}>T_{s} \end{array}\right.$$

The values of *S*_DS_ and *S*_D1_ in Equation (13) can be determined from the USGS website [42] for the 2475-year return period (2% probability of exceedance in 50 years), the 975-year return period (5% probability of exceedance in 50 years), and the 475-year return period (10% probability of exceedance in 50 years). After obtaining *S*_a_, the maximum equivalent seismic load *F*_equal_ for a bridge pier can be calculated. Consequently, the maximum seismic-induced moment demand *M*_equal_ at any section of a cantilever pier column can be calculated as follows:(14)$$M_{equal}=F_{\mathrm{equal}}\cdot L_{d}=(S_{a}\cdot W)\cdot L_{d}$$
where *L*_d_ is the distance from the equivalent seismic load point to the calculated section of the pier. Because the peak ground acceleration of an earthquake can be described by a Type II distribution [43], the spectral acceleration *S*_a_ also follows from Type II distribution. Accordingly, the seismic-induced moment demand that factors in uncertainty in Figure 3, determined by Equation (14), can be described by such a distribution as Equation (15):(15)$$F_{S\max}(s,T)=\exp[-{\left(b/s\right)}^{k}]$$
where *b* is the location parameter and *k* is the shape parameter; both parameters are site specific. According to Equations (11) and (15), the probabilistic distribution describing the individual seismic load demand shown in Figure 3 can be determined as follows:(16)$$F_{S}(s)=1-\frac{1}{\lambda \cdot T}\cdot {\left(b/s\right)}^{k}$$

To obtain parameters *b* and *k*, the respective acceleration response spectra for two annual frequencies of exceedance are determined by the USGS hazard curves [42] and the NCHRP response spectrum [41]. For example, the spectral response acceleration for 2% and 10% exceedance probabilities at the site location of a bridge are *S*_a1_ and *S*_a2_ from the acceleration response spectrum, respectively. By substituting these values into Equation (15), one obtains
(17)$$1-0.02=\exp[-{\left(b/(S_{a1}\cdot W\cdot L)\right)}^{k}]$$
(18)$$1-0.1=\exp[-{\left(b/(S_{a2}\cdot W\cdot L)\right)}^{k}]$$

By solving Equations (17) and (18), *b* and *k* can be determined. By substituting these two parameters into Equation (16), the distribution of individual seismic-induced moment demand *S* in Figure 3 is determined, and this distribution will be used in the following time-dependent reliability calculation process.

## 4. Time-Dependent Reliability 

In Figure 3, *n* independent earthquakes are supposed to occur within time interval $\left(0,t_{L}\right]$ at time instants *t*_j_, where *j* = 1,2,3…,*n*. Then, the probability that the bridge pier survives through *t*_L_ can be expressed by the reliability function [38]:(19)$$L(t_{L},n)=P\left[R(t_{1})>S_{1}\cap R(t_{2})>S_{2}\cdot \cdot \cdot \cap R(t_{n})>S_{n}\right]$$
where *R*(*t*_j_) and *S*_j_ are, respectively, the moment resistance and individual seismic-induced moment demand at time *t*_j_, which can be determined with the method in Section 2 and that in Section 3, respectively. According to the assumption that seismic events are independent and the seismic occurrence time is uniformly distributed in time interval $\left(0,t_{L}\right]$, Equation (19) can be further expressed as follows [38]:(20)$$L(t_{L},n)={\left[\frac{1}{t_{L}}\cdot \int_{0}^{t_{L}}F_{S}(R(t))dt\right]}^{n}$$
where *F*_S_ is the cumulative probability function described by Equation (16). Because the earthquake occurrence number *n* in Equation (20) is a random parameter, the reliability function should be calculated by using the theorem of total probability, as described by Equation (21):(21)$$\begin{array}{l} L(t_{L})=\sum_{n=1}^{\infty }L(t_{L},n)\cdot P(N(t_{L})=n)\\ =\sum_{n=1}^{\infty }{\left[\frac{1}{t_{L}}\cdot \int_{0}^{t_{L}}F_{S}(R(t))dt\right]}^{n}\cdot \frac{{\left(\lambda t_{L}\right)}^{n}}{n!}e^{-\lambda t_{L}} \end{array}$$

On the basis of Equations (8) and (21), Equation (22) is obtained:(22)$$L(t_{L})=\int_{0}^{\infty }\exp[-\lambda \cdot (t_{L}-\int_{0}^{t_{L}}F_{S}(r\cdot g(t))dt)]\cdot f_{R_{0}}(r)dr$$
where $f_{R_{0}}(r)$ is the probability density function of pristine pier’s moment resistance *R*_0_. From Equations (16) and (22), *L*(*t*_L_) further becomes
(23)$$L(t_{L})=\int_{0}^{\infty }\exp\left[-\frac{1}{T}\cdot {\left(\frac{b}{r}\right)}^{k}\cdot \int_{0}^{t_{L}}\frac{1}{{\left(g(t)\right)}^{k}}dt\right]\cdot f_{R_{0}}(r)dr$$

Then, the time-dependent failure probability of a given bridge pier section is calculated as
(24)$$P_{f}(t_{L})=1-L(t_{L})$$

From the above illustration, it is easily concluded that the failure probability of a corroded pier column is affected by both moment resistance *R*(*t*) and moment demand *S* at a critical section. For the coastal bridge column subjected to nonuniform corrosion, two failure modes should be considered. When the service time *t*_L_ and corrosion level is small, the potential failure section of a coastal bridge pier is at the pier bottom because this is the location of the largest bending-moment demand along the pier elevation. When service time *t*_L_ is long and corrosion damage is severe, the pier may first fail at the bottom of splash and tidal zone induced by the moment resistance deterioration. Therefore, the time-dependent reliability of a coastal bridge column can be expressed with the failure probability as
(25)$$P_{f}(t_{L})=\max(P_{f,\mathrm{sp}}(t_{L}),\;P_{f,\mathrm{sub}}(t_{L}))$$

where $P_{f,\mathrm{sub}}(t_{L})$ and $P_{f,\mathrm{sp}}(t_{L})$ are the sectional failure probability at the submerged zone and the splash and tidal zone, respectively. A flowchart to calculate the time-dependent reliability of the coastal bridge pier subjected to nonuniform corrosion is shown in Figure 5.

## 5. Time-Variant Reliability Analysis of Coastal Bridge Piers

A representative pier column of the California bridge inventory [44] is analyzed in the present study. The details of the column are presented in Figure 6. The pier has a clear height of 6.6 m, with a cross section size of 1829 × 914 mm. In the bridge pier, there are 36#11 longitudinal rebars with a mean nominal diameter of 35.81 mm. The corresponding steel reinforcement ratio is 2.07%. As for stirrups, the diameter and spacing are 16 mm and 300 mm, respectively. The randomness of the reinforcement and concrete material properties is factored in by using the probabilistic distribution adopted from the existing studies, as listed in Table 1. A mass of 485 tons was considered at the pier top, which represents the mass distribution from the superstructure.

The bridge pier is assumed to be in a coastal region in California with a latitude and longitude of 33.23 and −117.4, respectively. The site class is 760 m/s (B/C boundary). In order to study the influence of nonuniform corrosion characteristics on the time-dependent failure probability of the coastal bridge pier, three cases of corrosion-zone height distribution are considered. For case 1, the height of three zones, namely *L*_at_ (the height of the atmospheric zone), *L*_sp_ (the height of the splash and tidal zone), and *L*_sub_ (the height of the submerged zone), are 3.1 m, 3.5 m, and 0 m, respectively. For case 2, *L*_at_, *L*_sp_, and *L*_sub_ are 3.1 m, 2 m, and 1.5 m, respectively. For case 3, *L*_at_, *L*_sp_, and *L*_sub_ are 3.1 m, 1 m, and 2.5 m, respectively. The uncertainties of the corrosion parameters at the atmospheric zone and at the splash and tidal zone of the pier are described with the probabilistic distribution function, as listed in Table 1. The corrosion damage at the submerged zone is neglected owing to the deficiency of oxygen and light [39].

### 5.1. Time-Dependent Moment Resistance

To obtain the time-dependent moment resistance distribution of the bridge pier, a Monte Carlo simulation with 10,000 iterations is carried out. According to the simulation results, the time of corrosion initiation fits well into the lognormal distribution (Figure 7). As is shown in Figure 7a, the mean corrosion-initiation time in the splash and tidal zone is 14.2 years. with a standard deviation of 16.7 years. In the atmospheric zone, larger mean and standard deviation values of the corrosion-initiation time are observed, where the corresponding values are 40.67 and 57.17 years, respectively, as shown in Figure 7b. The inconsistent corrosion-initiation time between these two zones is attributed to the differences in chloride concentration, and it will eventually lead to nonuniform moment resistance degradation along the pier elevation.

After corrosion initiation, the residual yield strength and cross-sectional area of reinforcement in the pier reduces with the increase in service time, as shown in Figure 8 and Figure 9. In Figure 8, the mean value and standard deviation of the time-variant normalized cross section area (NCSA) are calculated. NCSA is defined as the area of reinforcing steel *A*(*t*) at time *t* normalized by the initial area *A*_0_: NCSA(*t*) = *A*(*t*)/*A*_0_. This figure illustrates that the cross-sectional area of reinforcing steel more seriously deteriorates in the splash and tidal zone than in the atmospheric zone, according to a comparison of the NCSA in these two zones. For example, the mean NCSA for the atmospheric zone is 0.90 at 60 years. The corresponding value decreases to 0.68 for the splash and tidal region. Figure 8 also depicts that the uncertainty of the NCSA at both corrosion regions increases with service time. This phenomenon can be attributed to the joint effects of the variability of the initial reinforcement area, the corrosion-initiation time, and the corrosion current density. Similar trends are also found for time-variant normalized yield strength (NYS). NYS is defined as the yield strength of reinforcement *f*_y_(t) at time *t* normalized by the initial yield strength *f*_y0_: NYS(*t*) = *f*_y_(t)/*f*_y0_. The mean value of NYS, mean plus standard deviation (Mean + Std) of NYS, and mean minus standard deviation (Mean-Std) of NYS are plotted in Figure 9. From Figure 9, it can be found that the NYS more rapidly decreases in the splash and tidal zone than in the atmospheric zone. After 100 years, the mean NYS for the atmospheric zone dropped by 9.2%, with a standard deviation of 8%. The corresponding values for the splash and tidal zone become 25.7% and 11%, respectively.

Similarly, the deterioration of concrete is determined with time-variant normalized concrete cover strength (NCCS). NCCS is defined as the concrete strength $f_{c}(t)$ at time *t* normalized by initial concrete strength $f_{c0}$: NCCS(*t*) = *f*_y_(t)/*f*_y0_. The mean value of NCCS, mean plus standard deviation (Mean + Std) of NCCS, and mean minus standard deviation (Mean-Std) of NCCS are plotted in Figure 10. From Figure 10a, it is found that the mean NCCS in the splash and tidal zone has dropped dramatically by 87.6% during the first 30 years, which can be attributed to the large probability of concrete cover spalling. Subsequently, the reduction rate of NCCS slows down, and its variability decreases with the increase in service time. By comparing Figure 10a with Figure 10b, it can be found that the reduction of NCCS in the atmospheric zone is less obvious than that of NCCS in the splash and tidal zone. This is due to the smaller mean and standard deviation of the corrosion-initiation time and the higher corrosion current density in the splash and tidal zone than in the atmosphere zone, as shown in Figure 7. After the 100-year service time, the residual mean and the standard deviation of NCCS in the atmospheric zone are 0.186 and 0.387, respectively. The corresponding values in the splash and tidal zone are 0.016 and 0.124, respectively.

After obtaining the probabilistic distribution of mechanical properties for the reinforcement and concrete, the M–N interaction diagrams of the pier for a time range of 100 years are determined through a Monte Carlo simulation. Then, the moment resistance with uncertainty under the axial force is obtained with the method in Section 2.3. The moment resistance data are further fitted into lognormal distribution. Take 0, 30, 50, 70, and 100 years as examples: the fitted probabilistic distribution of moment resistance is shown in Figure 11. From Figure 11a, it is found that there is an obvious decreasing trend of moment resistance for the splash and tidal zone with the increase in time. The mean moment resistance of the uncorroded column is 7962 kN⋅m; the corresponding value decreases by 18.36%, 29.51%, 38.48%, and 49.04% at 30, 50, 70, and 100 years, respectively. The deterioration of moment resistance at the atmospheric zone is less severe than that at the splash and tidal zone, according to a comparison of Figure 11a,b. At 30, 50, 70, and 100 years of the corroded pier, the mean moment resistance for the atmospheric zone increases by 14.91%, 26.1%, 37.64%, and 54.42% compared with that for the splash and tidal zone, respectively. This phenomenon is attributed to a smaller corrosion-initiation time and a more rapid corrosion propagation for the splash and tidal zone than those for the atmospheric zone.

Similar to Figure 11, the moment resistance distribution for every 10 years of the pier during the 100-year service period is determined. By calculating the mean value of sectional moment capacity at the atmospheric zone and the splash and tidal zone for every 10 years, the discrete values of *g*(*t*_i_) in Equation (9) are obtained. *g*(*t*_i_) is represented by circle and triangle symbols in Figure 12. These values can be fitted with the following moment resistance reduction function:(26)$$g(t)=1+\alpha_{1}\cdot t+\alpha_{2}\cdot t^{2}$$
where $\alpha_{1}=-7.06\times 10^{-3}$ and $\alpha_{2}=1.974\times 10^{-5}$ for the splash and tidal zone. For the atmospheric zone, $\alpha_{1}=-2.355\times 10^{-3}$ and $\alpha_{2}=-1.457\times 10^{-7}$. The moment resistance reduction function *g*(*t*) is also represented by blue and red lines in Figure 12. *g*(*t*) will be used for the reliability calculation. From Figure 12, it is found that *g*(*t*) for both corroded zones is fitted well with the parabolic function. Additionally, the splash and tidal zone experiences rapider resistance deterioration than the atmospheric zone during the service life of the pier. At 100 years, *g*(*t*) is 0.77 in the atmospheric zone. The corresponding value decreases to 0.51 in the splash and tidal zone.

### 5.2. Seismic-Induced Moment Demand

According to the USGS mapping project and NCHRP response spectrum, as mentioned in Section 3, the respective acceleration response spectra corresponding to two seismic exceedance probabilities are established, as shown in Figure 13. In this figure, the 10% in 50 years corresponds to an earthquake return period of 475 years. The 2% in 50 years corresponds to a return period of 2475 years. A preliminary calculation shows that the time-variant period of the coastal bridge pier ranges from 0.81 to 0.92 s. In this period range, the spectral response acceleration nonlinearly decreases with the increase in the period. Therefore, from a conservative viewpoint, the spectral response acceleration corresponding to the period of 0.81 s is adopted during the analysis. Based on Figure 13, the spectral response acceleration at 0.81 s is calculated to be 0.399 and 0.185 g for 2% and 10% exceedance probabilities in 50 years, respectively. Incorporating these two acceleration values into Equations (15) and (16), the value of parameters *b* and *k* are calculated to be 2078 and 2.149, respectively. Thus, the cumulative probability distribution of individual seismic demand can be determined by Equation (16) with a mean earthquake occurrence rate of λ=2/year. The adoption of λ=2/year is based on the study in [50], which indicates that some specific regions of California have a high probability of 1.5 to 2.0 earthquakes in a year.

On the basis of Equations (15) and (16), the cumulative probability of individual seismic demand and maximum seismic demand in 50 years is determined, as shown in Figure 14. It is found that the moment demand induced by the maximum seismic load in 50 years is significantly larger than that by an individual seismic load under the same cumulative probability. This is because the probability of greater earthquake intensity (PGA) increases with the increase in the pier’s service time. Such a phenomenon is one of the reasons that lead to the variation in the seismic reliability of coastal bridge pier with time.

### 5.3. Time-Dependent Reliability Analysis

According to the above-determined time-dependent moment resistance distribution and moment demand distribution induced by the seismic load, the sectional failure probability along the pier elevation for three different corrosion cases at some service years can be calculated with Equation (24), as shown in Figure 15.

For case 1, as presented in Figure 15a, it is found that the failure probability of the pier increases from the pier top to the pier bottom for the same year thanks to the increase in the bending moment. Because of the corrosion of reinforcement, there is a significant change in failure probability at the intersection of two corrosion regions. Figure 15a also shows that the failure probability of the pier is increasing with the increase in service time for the same section, owing to corrosion damage. For instance, the failure probability of the bottom section is 0.044 at 30 years. The corresponding value increases to 0.081, 0.138, and 0.242 at 50, 70, and 100 years, respectively.

Compared with case 1, the failure probability of the pier in case 2 is different, as shown in Figure 15b. From this figure, it is found that the critical section of the pier shifts from the pier bottom to the splash and tidal zone during the service period. When the service times are 30 and 50 years, the critical section is located at the pier bottom owing to maximum moment demand and not severe corrosion damage of the splash and tidal zone. When the service time of the pier reaches 100 years, the critical section shifts to the bottom of the splash and tidal zone. At this year, the failure probability of the pier bottom is 0.114, and the corresponding value is 0.143 at the bottom of the splash and tidal zone. With the submerged zone increased to 2.5 m, as shown in Figure 15c, the failure probability along the pier height of case 3 is similar to that of case 2. However, the critical section is in the pier bottom and does not shift to the splash and tidal zone during the 100 years.

To further investigate the correlation between corrosion region distribution and the pier failure modes, the time-dependent sectional failure probability of two potential critical sections is selected and presented in Figure 16. For case 1 (the bottom of the splash and tidal zone is the pier bottom), the time-dependent failure probability of the bottom section is compared between the condition with and the condition without corrosion, as shown in Figure 16a. From this figure, it is found that the failure probability is almost linearly increased with the increase in service time for the pier without corrosion at the splash and tidal zone. This phenomenon reflects the increase in maximum seismic demand with the increase in service time. When the pier is corroded in the splash and tidal zone, the coupling effect of corrosion damage and increased seismic demand leads to a nonlinear rise in the failure probability, which is significantly larger than that without the corrosion condition. For example, when the service time is 100 years, the failure probability at the pier bottom for the uncorroded condition is 0.114, and the corresponding value increases to 0.244 for the corrosion condition. For case 2, as shown in Figure 16b, there is an intersection point between two curves at about 70 years. This phenomenon means that the failure section shifts from the pier bottom to the bottom of the splash and tidal zone at 70 years. For case 3 (Figure 16c), the failure probability at the bottom of the splash and tidal zone is always smaller than that of the pier bottom. The failure mode will not change during the 100 service years.

On the basis of the sectional failure probability, the time-dependent reliability of the whole pier expressed with failure probability can be determined with Equation (25), which is shown in Figure 17. To give the gradual evolutionary trend of the time-dependent failure probability curve, the cases of submerged zone height *L*_sub_ = 1.0 m and 2.0 m are also calculated and plotted in this figure. It is observed that not only service time but also submerged zone height influence the failure probability of the pier. Additionally, the increase in *L*_sub_ delays the time when the failure section shifts from the pier bottom to the bottom of the splash and tidal zone. For the cases where *L*_sub_ increases from 1.0 to 1.5 and 2.0 m, the year of critical section shift increases from about 50 to 70 and 90 years. When *L*_sub_ is 2.5 m, the shift year is delayed beyond the service year and the failure probability is the same as the pristine pier. From Figure 17, it is also observed that the failure probability of coastal bridge pier decreases with the increase in the submerged zone. At the end of its service life, the failure probability for *L*_sub_ = 0 m is 0.244, and the corresponding value decreases to 0.178, 0.148, 0.12, and 0.11 for *L*_sub_ = 1.0, 1.5, 2.0, and 2.5 m, respectively.

## 6. Previous Research into Time-Dependent Seismic Reliability of Corroded Bridge Piers

Realizing the adverse effect of corrosion on the seismic performance of RC components, some researchers assessed the seismic reliability of corroded bridge piers [27,31,32]. However, the method of obtaining the moment resistance of bridge piers with time-dependent M–N interaction diagrams and determining the seismic demand with acceleration response spectrum is seldomly used. This section presents the comparable previous work [27] into the seismic reliability evaluation of corroded bridge piers to highlight the limitations of the work.

Ref. [27] proposed a new deterministic corrosion rate model and then used this model in assessing the time-dependent seismic reliability of corroded bridge piers. They considered that the distribution of random variables for both corrosion environment and material properties in moment resistance calculation is uniform along the pier height. This assumption is not applicable to the nonuniform sectional moment resistance calculation along the elevation for the coastal bridge piers. They considered the seismic demand as constant by using the design response spectrum. Therefore, the randomness of a seismic attack was not included in the seismic reliability analysis of bridge piers.

## 7. Conclusions

In this study, a method to estimate the time-dependent reliability of coastal bridge piers subjected to nonuniform corrosion was proposed. The time-dependent moment resistance of the pier was established using the Monte Carlo simulation, and the randomness of seismic-induced moment demand was treated as a Poisson point process. Using the theorem of total probability, the formulas of the time-dependent failure probability of nonuniform corroded bridge piers are derived. As an illustration of the proposed method, a case study example was analyzed. Some conclusions are summarized as follows:Owing to the more severe corrosion effect in the splash and tidal zone, the corrosion-initiation time, time-varying NCSA, NYS, and NCCS in this zone are smaller than those in the atmospheric zone. At the 100-year service time, the respective mean values of NYS and NCCS in the atmospheric zone drop by 9.2% and 81.4% compared with that at 0 years. The corresponding reduction percentage reaches up to 25.7% and 98.4% in the splash and tidal zone.The time-dependent failure probability for uncorroded pier sections is almost linearly increased with time thanks to the reason that maximum seismic demand increases with time. For the corroded pier section in the splash and tidal zone, the corrosion damage and the increase in seismic demand lead to a nonlinear increase in failure probability.Different from the uniform corroded urban bridge pier, whose failure occurs at the bottom section, the failure mode of the coastal bridge pier is not singular and is determined by the service time and the distribution of nonuniform corroded zones. The increase in the submerged zone height delays the year when the failure section transfers from the pier bottom to the bottom of the splash and tidal zone of the whole pier.The position of the submerged zone significantly influences the time-dependent failure probability of the coastal pier element. At the end of the pier’s service life, the pier’s failure probability decreases from 0.244 to 0.178, 0.148, 0.12, and 0.11 when the submerged zone height increases from 0 to 1.0, 1.5, 2.0, and 2.5 m, respectively.

## Figures and Tables

**Figure 1 materials-16-01029-f001:**
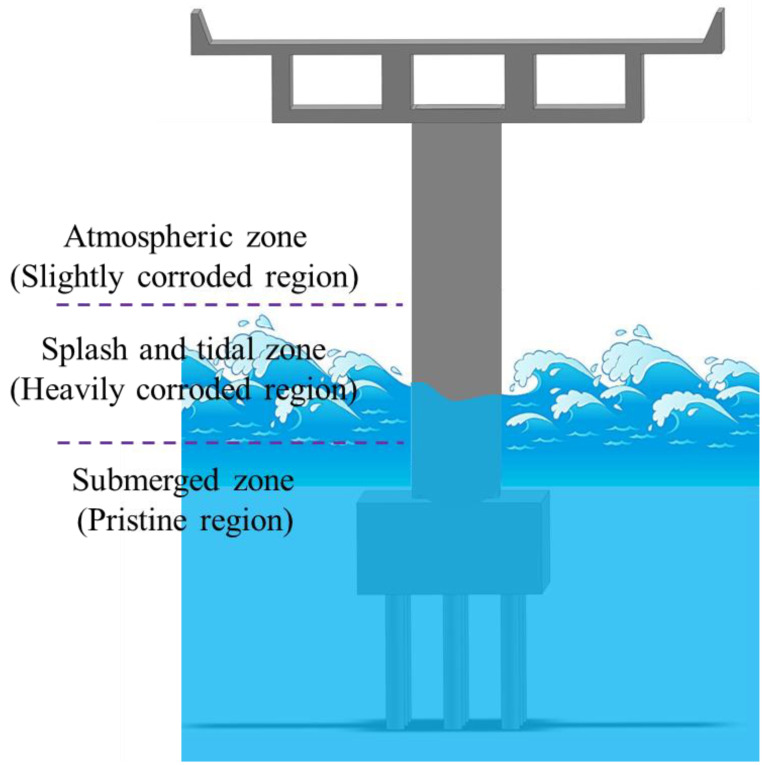
Schematic of the coastal bridge pier subjected to nonuniform corrosion.

**Figure 2 materials-16-01029-f002:**
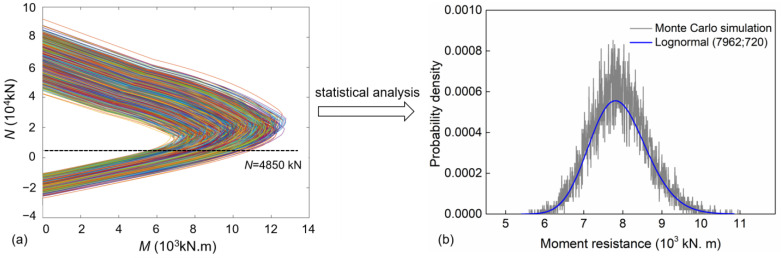
Moment resistance distribution of a pier, based on an M–N interaction diagram. (**a**) calculated M-N interaction diagrams (**b**) Probability distribution of moment resistance.

**Figure 3 materials-16-01029-f003:**
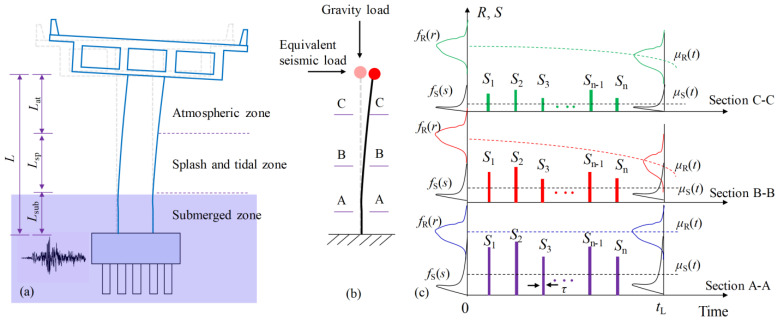
Schematic of seismic load action and the time-dependent resistance for a pier. (**a**) non-uniform corroded pier (**b**) equivalent force model (c) seismic load demand and resistance.

**Figure 4 materials-16-01029-f004:**
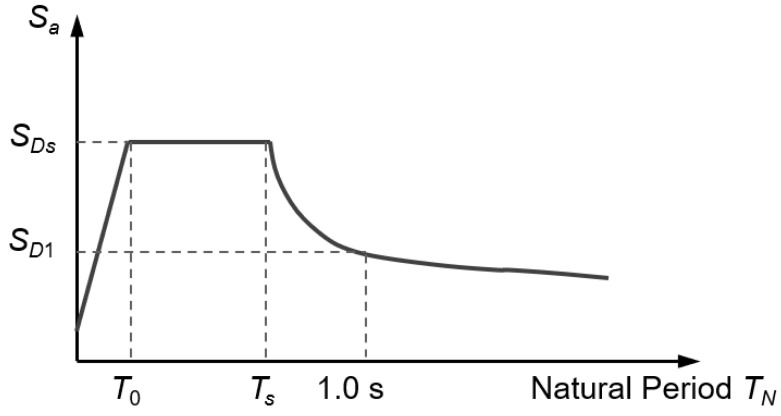
Response spectrum.

**Figure 5 materials-16-01029-f005:**
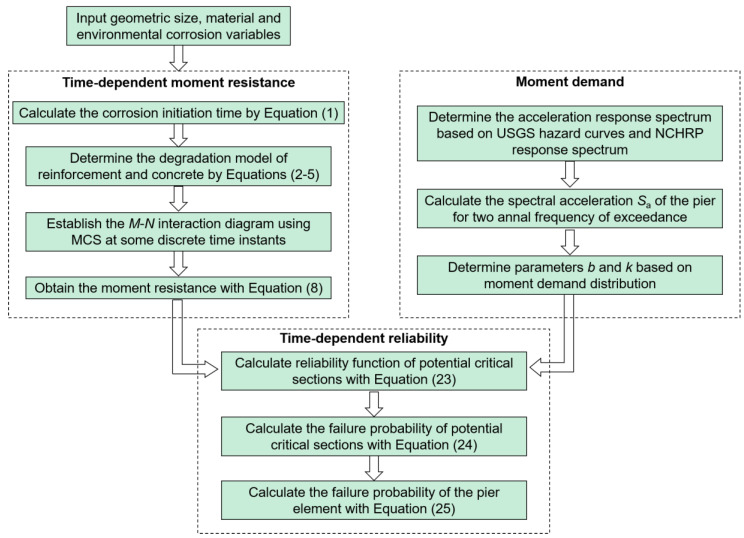
Flowchart of time-dependent reliability calculation.

**Figure 6 materials-16-01029-f006:**
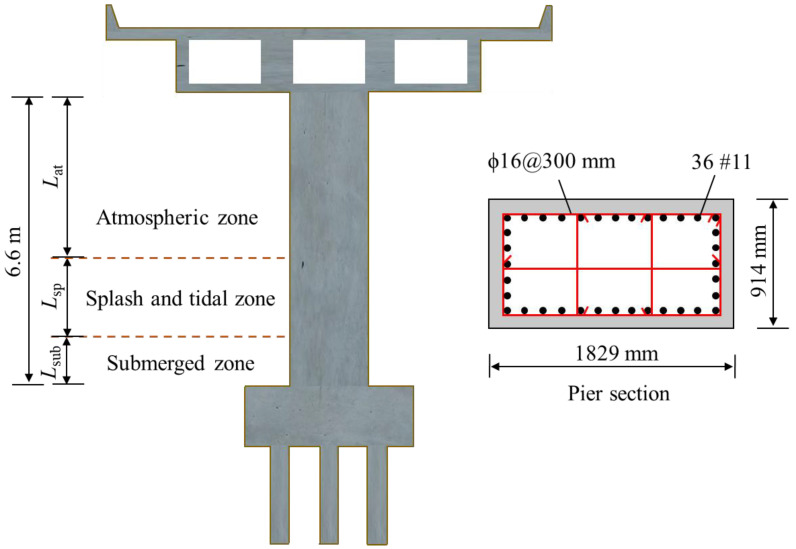
Schematic diagram of the coastal bridge pier.

**Figure 7 materials-16-01029-f007:**
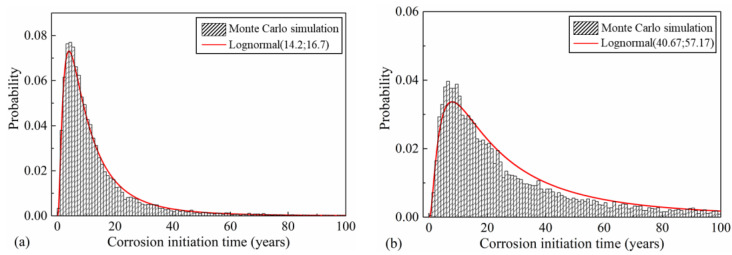
Distribution of corrosion-initiation time: (**a**) splash and tidal zone; (**b**) atmospheric zone.

**Figure 8 materials-16-01029-f008:**
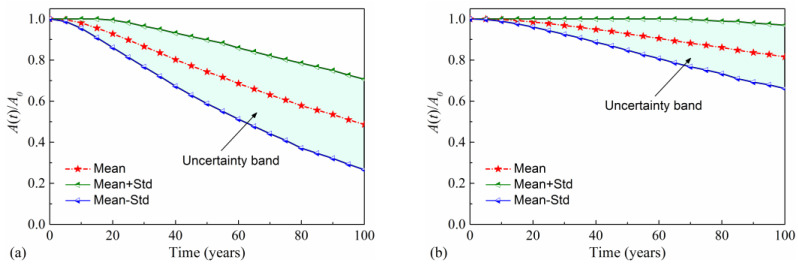
Time-variant NCSA of reinforcement: (**a**) splash and tidal zone; (**b**) atmospheric zone.

**Figure 9 materials-16-01029-f009:**
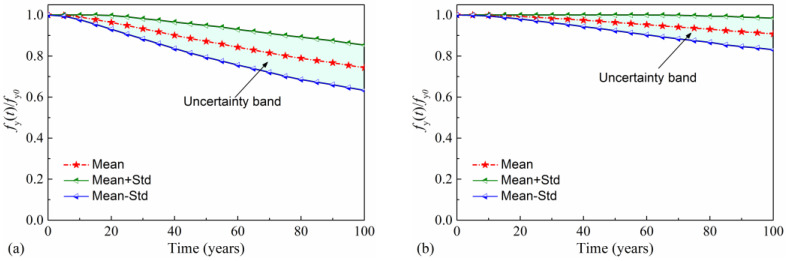
Time-variant NYS of reinforcement: (**a**) splash and tidal zone; (**b**) atmospheric zone.

**Figure 10 materials-16-01029-f010:**
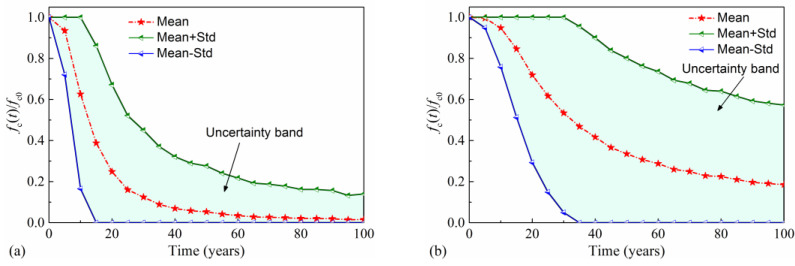
Time-variant NCCS of concrete: (**a**) splash and tidal zone; (**b**) atmospheric zone.

**Figure 11 materials-16-01029-f011:**
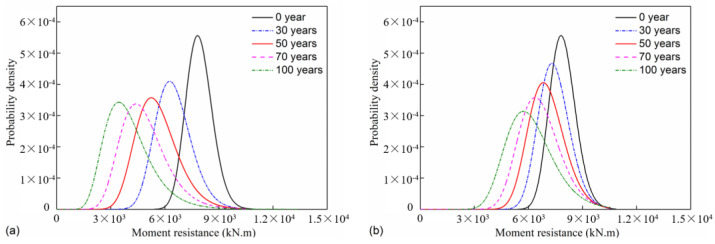
Moment resistance distribution: (**a**) splash and tidal zone; (**b**) atmospheric zone.

**Figure 12 materials-16-01029-f012:**
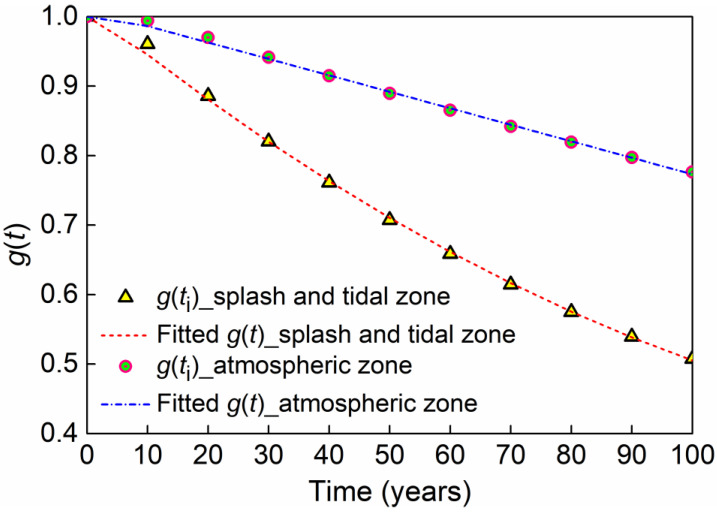
Moment resistance reduction function.

**Figure 13 materials-16-01029-f013:**
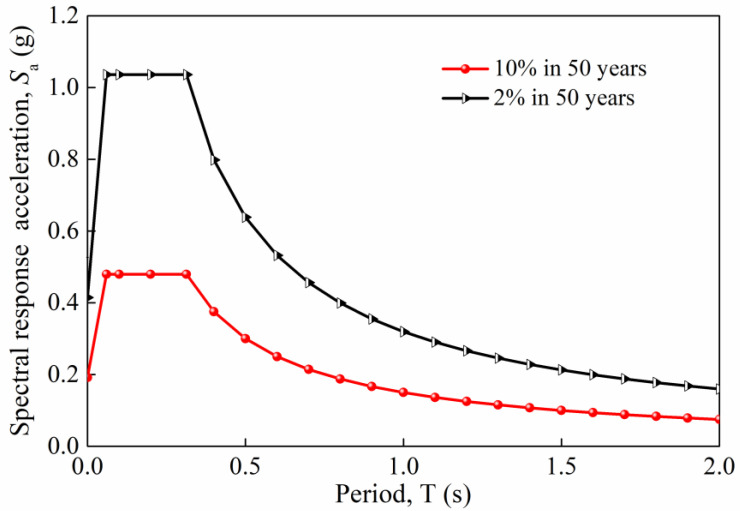
Acceleration response spectrum for two seismic exceedance possibilities.

**Figure 14 materials-16-01029-f014:**
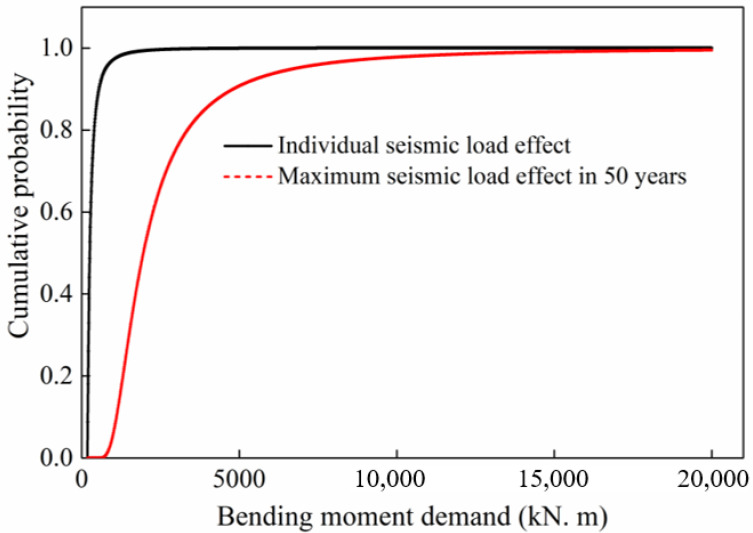
Cumulative probability of seismic load demand.

**Figure 15 materials-16-01029-f015:**
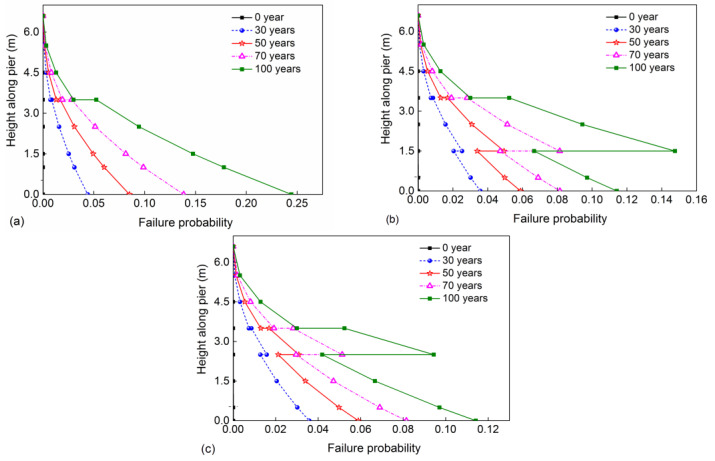
Sectional failure probability along the pier elevation: (**a**) case 1; (**b**) case 2; (**c**) case 3.

**Figure 16 materials-16-01029-f016:**
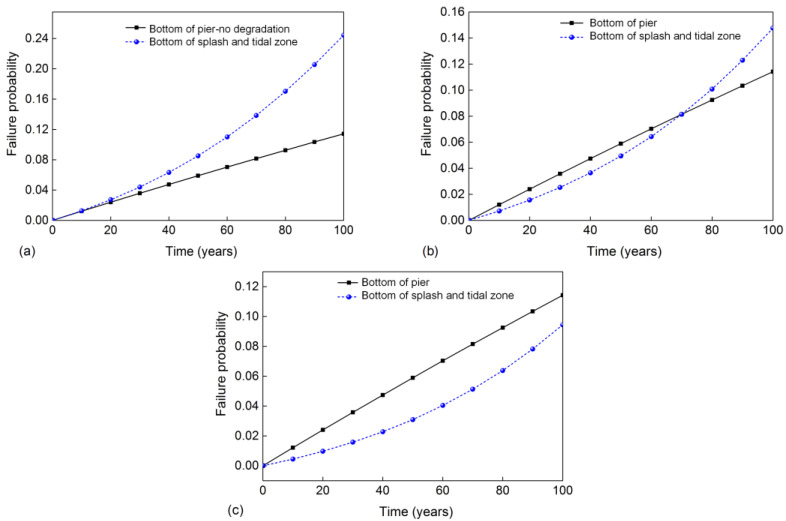
Time-dependent failure probability of pier sections: (**a**) case 1; (**b**) case 2; (**c**) case 3.

**Figure 17 materials-16-01029-f017:**
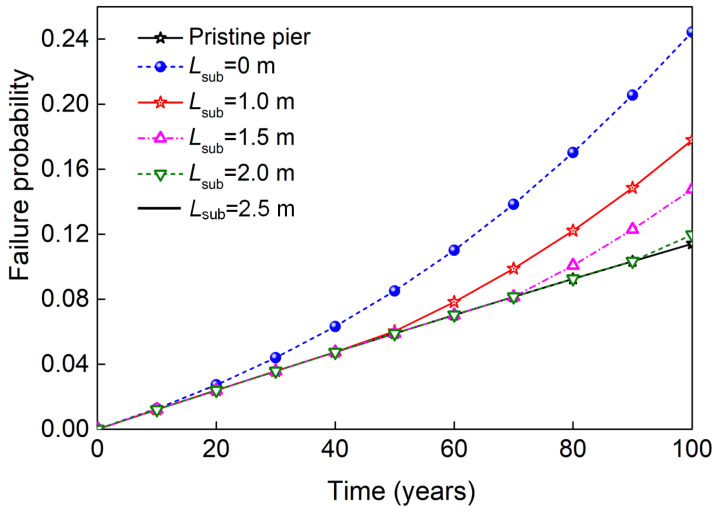
Time-dependent failure probability of the whole pier.

**Table 1 materials-16-01029-t001:** Random variables for the reliability analysis of the coastal bridge pier.

Variables	Distribution			Source
	Type	μ	CV	
*c* (mm)	Normal	60	0.16	[45]
*f*_y0_ (MPa)*f*_c0_ (MPa)	LognormalNormal	46529.03	0.0800.124	[46]
*D*_0_ (mm)	Normal	35.81	0.02	[47]
*D*_c_ (mm^2^/year)	Lognormal	124	0.7	[48]
*C*_0___splash_ (kg/m^3^)	Lognormal	7.35	0.7	[49]
*C*_0___atmos_ (kg/m^3^)	Lognormal	2.95	0.7	[49]
*C*_cr_ (kg/m^3^)	Uniform	0.9	0.19	[8]
*i*_corr_atmos_ (μA/cm^2^)	Lognormal	2.586	0.667	[35]
*i*_corr_splash_ (μA/cm^2^)	Lognormal	6.035	0.571	[35]

Notes: The subscripts splash and atmos in the variables represent the splash and tidal zone and the atmospheric zone, respectively. *μ* and CV denote the mean and the coefficient of variation of the distribution, respectively.

## Data Availability

Not applicable.

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
