# Peer review of "Time-Dependent Seismic Reliability of Coastal Bridge Piers Subjected to Nonuniform Corrosion"

_materials, 2023, doi:10.3390/ma16031029_

Round 1

Reviewer 1 Report

The papers subject is interesting and important to engineers; however, it needs to carefully address the following comments.

Comments:

1. Please rewrite the abstract.

2. A additional correlation with DIANA or similar FEA program would assist in assuring the conclusions are correct.

3. Please rewrite the conclusions to be more quantitatively.

Author Response

The response to the reviewer's comments is uploaded in a Word file

Reviewer 2 Report

The manuscript presents an analytical analysis of how non-uniform corrosion affects the seismic behavior of coastal bridge piers. Generally, the paper was organized very well and the results were well analyzed and reviewed. The paper is certainly worth publishing after considering the following comments:

Comment 1: Lines 72-73: Please revise the sentence.

Comment 2: Eqs. 2 and 4: The authors should explain how the values of icorr and Dt are calculated and obtained.

Comment 3: Eq. 5: What is the value of K?

Comment 4: Lines 154-156: Please revise the sentence.

Comment 5: Citing recent articles is also recommended. The most recent article is from 2019.

Author Response

The response to the reviewer's comments is uploaded as a Word file

Reviewer 3 Report

SUMMARY

The article submitted for review is relevant. It considers the seismic reliability of coastal bridge piers subjected to non-uniform corrosion. The relevance lies in the fact that coastal bridge supports are subject to random degradation of performance due to the presence of complex heterogeneous corrosion manifestations and uncertainties of the material itself. These uncertainties and manifestations must be accompanied by impairments, predictions and controlled by reliability assessment methods. The study provides a new methodology for calculating the reliability of a coastal bridge support depending on time, taking into account a sudden seismic event and uneven corrosion. Modern methods are used, such as the Monte Carlo simulation method, the Poisson model. Finally, formulas are derived for calculating, depending on time, the reliability of an inhomogeneous corrosion pile. A number of important results have been established, which makes the article practically significant and scientifically new. However, it does have some drawbacks, which are listed below.

COMMENTS

1.     The abstract is somewhat unstructured. It should consist in highlighting a scientific problem. Its wording should be clear, that is, if the authors talk about random manifestations of seismic, then this reason must be indicated. The authors then talk about corrosion manifestations and material uncertainty. In the beginning, they don't talk about seismic, but only about corrosion and heterogeneity of the material. Thus, the authors have some inconsistencies within the abstract itself, which needs to be brought into uniformity.

2.     As far as it is clear from the abstract, there are three problems, and no one has yet approached their study in such a complex way. You need to understand what the specific scientific novelty is. It remains unclear whether this is the proposed approach or the fact that known methods are applied to a new problem that no one has dealt with before. The authors should specify this point, because the rest of the article is repelled from it. In addition, the quantitative aspect of the result obtained is unclear. Perhaps the authors will be able to give at least one phrase a comparative aspect, which makes their study better than previously known ones.

3.     The Introduction should be strengthened in terms of the analytical component. The authors list 25 sources on the topic of the study, while some of them are considered very superficially. Authors need to do an in-depth review of the literature.

4.     Section 2, in principle, is presented quite interestingly and in detail, but I would like to see explanations for Figure 2. The only remark to the authors will be the need to indicate the materials used. That is, the authors should place the considered and applied materials in a separate section. Since the Journal to which the authors submit their article is called "Materials", more attention should be paid to the concrete itself and its heterogeneity. However, the authors pay more attention to methodology and mathematical models, while somewhat omitting the material science component.

5.     Differences between Figure 1 and Figure 5 are not clear. There is not enough explanation for the scale presented in Figure 5. The second part of Figure 5 looks unreadable, it must be presented either as a separate figure or in a higher resolution. The same remark applies to Figures 4 and 6. The graphs in Figures 8-11 need more explanation. The seismic model in Figures 12 and 13 seems interesting, but unfortunately they are presented in poor quality and not explained.

6.     The authors presented an inconclusive comparison of their results with those of other authors. Discussions should include more detailed analysis, emphasizing scientific novelty.

7.     The “Conclusions” section needs to be improved in terms of determining the practical prospects for research and concretizing the scientific result: whether new knowledge was obtained, existing ideas were developed, that is, what was new for science.

8.     For such an actual topic, 39 sources are very few. Perhaps the number of analyzed literature should be increased to 50 sources. At the same time, it is important to give preference to sources for the last 5 years.

9.     In general, the reviewer evaluates the article positively, however, there are doubts about the correspondence of its content to the Journal "Materials". Maybe the authors should choose another Journal that is more focused on mathematical models and calculations? But this remark is not critical, since this article is very informative from an engineering point of view.

Author Response

The response to the reviewer's comments is uploaded as a Word file.

Round 2

Reviewer 3 Report

The authors noted all comments. The paper has been greatly improved and is now available for publication.